# Acute Kidney Injury and In-Hospital Mortality: A Retrospective Analysis of a Nationwide Administrative Database of Elderly Subjects in Italy

**DOI:** 10.3390/jcm8091371

**Published:** 2019-09-02

**Authors:** Fabio Fabbian, Caterina Savriè, Alfredo De Giorgi, Rosaria Cappadona, Emanuele Di Simone, Benedetta Boari, Alda Storari, Massimo Gallerani, Roberto Manfredini

**Affiliations:** 1Faculty of Medicine, Surgery and Prevention, University of Ferrara, via Ludovico Ariosto 35, 44121 Ferrara, Italy (C.S.) (R.C.) (R.M.); 2Azienda Ospedaliero-Universitaria ‘S. Anna’, Via Aldo Moro 8, 44123 Cona, Ferrara, Italy (A.D.G.) (E.D.S.) (B.B.) (A.S.) (M.G.)

**Keywords:** acute kidney injury, in-hospital mortality, comorbidity, International Classification of Diseases, 9th Revision, Clinical Modification (ICD-9-CM)

## Abstract

Background: The aim of this study was to investigate the association between acute kidney injury (AKI) and in-hospital mortality (IHM) in a large nationwide cohort of elderly subjects in Italy. Methods: We analyzed the hospitalization data of all patients aged ≥65 years, who were discharged with a diagnosis of AKI, which was identified by the presence of the International Classification of Diseases, 9th Revision, Clinical Modification (ICD-9-CM), and extracted from the Italian Health Ministry database (January 2000 to December 2015). Data regarding age, gender, dialysis treatment, and comorbidity, including the development of sepsis, were also collected. Results: We evaluated 760,664 hospitalizations, the mean age was 80.5 ± 7.8 years, males represented 52.2% of the population, and 9% underwent dialysis treatment. IHM was 27.7% (210,661 admissions): Deceased patients were more likely to be older, undergoing dialysis treatment, and to be sicker than the survivors. The population was classified on the basis of tertiles of comorbidity score (the first group 7.48 ± 1.99, the second 13.67 ± 2,04, and third 22.12 ± 4.13). IHM was higher in the third tertile, whilst dialysis-dependent AKI was highest in the first. Dialysis-dependent AKI was associated with an odds ratios (OR) of 2.721; 95% confidence interval (CI) 2.676–2.766; *p* < 0.001, development of sepsis was associated with an OR of 1.990; 95% CI 1.948–2.033; *p* < 0.001, the second tertile of comorbidity was associated with an OR of 1.750; 95% CI 1.726–1.774; *p* < 0.001, and the third tertile of comorbidity was associated with an OR of 2.522; 95% CI 2.486–2.559; *p* < 0.001. Conclusions: In elderly subjects with AKI discharge codes, IHM is a frequent complication affecting more than a quarter of the investigated population. The increasing burden of comorbidity, dialysis-dependent AKI, and sepsis are the major risk factors.

## 1. Introduction

Comorbidity is a risk factor for in-hospital mortality (IHM) and appears to be high in admitted elderly patients [1]. In elderly subjects, admissions due to acute medical complications, determinants of health status decline, and prediction of negative outcomes could be based on comorbidities [2]. Mortality has been associated with the development of acute kidney injury (AKI) [3]. In western society, multi-morbidity is frequently and increasingly reported, causing increased complexity in elderly subjects with different burdens of organ dysfunction [4,5,6]. The presence of comorbidities could increase the risk of AKI, and in previous studies, our group, utilizing administrative databases, evaluated the relationship between renal dysfunction, comorbidities, and IHM in individuals hospitalized with myocardial infarction, pulmonary embolism, stroke, and severe chronic obstructive pulmonary disease [7,8,9,10]. Even a small increase in serum creatinine during admission has been significantly associated with mortality, hospital length-of-stay (LOS), and costs, after adjustment for age, gender, severity of illness, and chronic kidney disease in a sample of 19,982 adults [11]. It has been recently reported that in a large USA database, AKI increased hospitalization costs and LOS, especially if dialysis was required [12]. More than 50% of elderly adults have three or more chronic diseases and poor general conditions are associated with many adverse consequences. A comorbidity index is able to summarize all the coexistent illnesses in a single numeric score, allowing comparisons between different groups—the index suggests the severity of the patient’s conditions [13]. In clinical practice, the Elixhauser Index is a well-known score able to summarize comorbidity into an index providing a single parameter [14]; however, because it was conceived in 1998, our group decided to modify and adapt it for our hospitalized patients [1]. Management of a specific disease in the presence of comorbidity is complex [15], and recognition of the high risk related to AKI in an aging population may help to avoid further morbidity and mortality. Elderly subjects are often treated similarly to the general population regardless of the comorbidity; this practice is unrealistic due to the fact that survival in sexagenarians is significantly different to survival in nonagenarians [16,17]. The risk factors for AKI in older adults are multiple comorbidities and polypharmacy [18,19], and AKI is a common complication in hospitalized older individuals [20]. The aim of this retrospective study was to evaluate the relationship between AKI, comorbidities, dialysis treatment, and IHM in a large sample of Italian elderly subjects.

## 2. Experimental Section

### 2.1. Patient Selection and Eligibility

The total number of hospitalizations due to AKI between 1 January 2000, and 31 December 2015, recorded in the Italian National Hospital database, provided by the Ministry of Health (SDO Database, Ministry of Health, General Directorate for Health Planning) were considered in this analysis. All hospitalizations of patients in public and private Italian hospitals are recorded in the National database of Hospital Discharge Records (HDR). Gender, age, date of hospital admission and discharge, department of admission and discharge, vital status at discharge (in-hospital death vs. discharged alive), main diagnosis, up to five co-morbidities, and up to six procedures/interventions, based on the International Classification of Diseases, 9th Revision, Clinical Modification (ICD-9-CM), are recorded in the HDR file. Because of the national disposition-by-law in terms of privacy, patients’ names and all other potential identifiers were removed by the Ministry of Health from the database for this analysis. The only identifier, allowing us to search the database for possible recurrence of hospital admissions of the same patient, was a consecutive number for each patient. Administrative database codes usually make reference to acute renal failure, but in clinical settings, the term AKI has largely replaced that term. AKI was identified using as first or second discharge diagnosis ICD-9-CM code 584.xx. Finally, diagnosis of sepsis and diabetes were taken into consideration, and LOS was calculated.

### 2.2. Data Analysis

The outcome of this analysis was IHM being a hard clinical outcome indicator; therefore, fatal cases (death during hospitalization) were compared with non-fatal ones (patient discharged alive). The comorbidity score was evaluated using ICD-9-CM, and a novel score recently proposed by our group was calculated for considering the comorbidity burden [1]. The conditions of age, gender, presence of chronic kidney disease, neurological disorders, lymphoma, solid tumor with metastasis, ischemic heart disease, congestive heart disease, coagulopathy, fluid and electrolyte disorders, liver disease, weight loss, and metastatic cancer were taken into account for score calculation. The original score was corrected, removing the diagnosis of previous AKI, therefore points assigned to renal diseases were considered only if chronic kidney disease was recorded during admission. The points for each condition ranged from 0 to 16, and the total score calculated could vary between 0 and 89. When the score was >40, the risk of IHM was >60%. The score, developed using administrative data, was calculated automatically. The points assigned to different conditions in order to calculate the risk score of IHM are reported in Table 1 [1]. Finally, data of IHM related to these patients were extracted from the general database.

### 2.3. Statistical Analysis

Absolute numbers, percentages, and means ± SD were used to present data. A descriptive analysis of the whole population was performed, followed by a comparison of survivors and deceased during admission. The population was divided in tertiles based on the comorbidity score, and the three groups were compared according to all the parameters investigated. The analysis of the variables was done by using the Chi-Squared test, Student *t*-tests, Mann–Whitney *U* test, and ANOVA as appropriate. Moreover, in order to assess the independent parameters associated with IHM, the latter was considered as a dependent variable in a logistic regression analysis and demography; tertiles of the comorbidity score, dialysis treatment, and sepsis were considered as independent ones. Odds ratios (ORs) and their 95% confidence intervals (95% CI) were reported. All *p*-values were 2-tailed, and *p*-value < 0.5 was considered significant. SPSS 13.0 for Windows (SPSS IN., Chicago, IL, USA, 2004) was used for statistical analysis.

### 2.4. Ethical Issues

This retrospective study was conducted in agreement with the declaration of Helsinki of 1975, revised in 2013. Subject identifiers were deleted before the analysis of the database in order to maintain data anonymity and confidentiality. None of the patients could be identified, either in this paper or in the database. The study was conducted in agreement with the existent Italian disposition-by-law (G.U. n.76, 31/03/2008), and due to the study design ethics committee approval was not necessary.

## 3. Results

We analyzed 760,664 records, of which 52.2% were males. The mean age of the population was 80.5 ± 7.8 years, and mean LOS was 13.72 ± 15.49 days. The mean comorbidity score was 14.57 ± 6.21. The distribution of the score in the population is shown in Figure 1.

The main characteristics of the investigated population are reported in Table 2.

Heart failure was diagnosed in 146,359 (19.2%), fluid and electrolyte disorders in 126,702 (16.7%), chronic kidney disease in 115,238 (15.1%), cancer in 71,624 (9.4%,), ischemic heart diseases in 34,943 (4.6%), liver diseases in 34,015 (4.5%), metastatic cancer in 30,425 (4%), neurological disorders in 17,912 (2.4%), lymphoma in 16,443 (2.2%), cachexia in 15,363 (2%), and coagulopathy in 8266 (1.1%). Diabetics were 15.6% (n = 118,299) and patients with diagnosis of sepsis 5.1% (n = 39,144). Nine percent of the population underwent dialysis treatment, and overall IHM was 27.7% (210,661 hospitalizations). The frequency of survivors and deceased patients during the study period is reported in Figure 2.

Deceased patients were older and the prevalence of women was higher. Moreover, deceased subjects had higher rates of sepsis diagnosis, were more likely to be on dialysis treatment, and were sicker than survivors; however, diabetes was more frequent in survivors (Table 3).

The comorbidity score of the population was classified on the basis of tertiles: in the first tertile, the score was lower than 11; in the second, it was between 11 and 17; and in the third, it was higher than 17. Comparison between the three groups is reported in Table 4.

IHM was independently associated with dialysis-dependent AKI (OR 2.721; 95% CI 2.676–2.766; *p* < 0.001), development of sepsis (OR 1.990; 95% CI 1.948–2.033; *p* < 0.001), the second tertile of comorbidity (OR 1.750; 95% CI 1.726–1.774; *p* < 0.001), and the third tertile of comorbidity (OR 2.522; 95% CI 2.486–2.559; *p* < 0.001) (Figure 3, the first tertile was used as the reference).

## 4. Discussion

To the best of our knowledge, this is the first national Italian study testing the impact of comorbidity on IHM in elderly subjects admitted with AKI. The score tested had an approximately normal distribution, and the population showed a gradual increase in the number of admissions and IHM during the study period. The mean score in the whole population was 14.57, and the mean score in deceased subjects was 15.96. However, when the tertiles were analyzed, in the highest tertile, we detected a mean score value of 22.12, and more than one-third of subjects belonging to this group died during hospitalization. Univariate analysis showed that the group of patients in the third tertile had a higher comorbidity burden due to a higher prevalence of cancer and cardiovascular morbidity, the two major causes of death in western societies. Such a burden of comorbidity was not related to sepsis or to dialysis-dependent AKI. Our data suggest that physicians did not think that dialyzing AKI patients with a high comorbidity burden could be beneficial. Probably, as a result of this finding, LOS in patients with a high comorbidity score was lower. Previously, we reported that the score used in this study and sepsis diagnosis were able to predict IHM in patients hospitalized for infectious diseases [21]. In this study, sepsis had a lower prevalence in the third tertile of the comorbidity burden; however, as well as increasing comorbidity, it was independently associated with IHM. In this study, a diagnosis of sepsis was made more frequently in the group of patients with a lower comorbidity burden, and in the same group, dialysis-dependent AKI had a higher prevalence, suggesting that there was a relationship between the two findings, i.e., subjects with sepsis developed dialysis-dependent AKI. Diabetes, a well-known risk factor for mortality, was recorded in 15.6% of cases, and appeared to be more frequent in survivors. However, this finding could be ascribed to a bias associated with our study design. Conditions pre-existing at the time of admission, such as diabetes, are poorly coded if the hospitalization procedure is complicated and if the patient dies [22]. In studies analyzing administrative databases, diabetes usually appears to be protective. The comparison of survivors and deceased patients could not show the real difference in the comorbidity burden, but the analysis of tertiles of the comorbidity burden and logistic regression analysis suggested that the three factors related to IHM were increasing comorbidity score due to cardiovascular and liver disease, cachexia and cancer, diagnosis of sepsis, and advanced renal damage requiring dialysis treatment. Increasing the tertiles of comorbidity score had an impact on mortality similar to dialysis-dependent AKI and sepsis.

We run the Hosmer–Lemeshow test in order to evaluate the goodness of fit of our logistic regression, and we could detect that our data were not fitting well into the model. The Hosmer–Lemeshow test calculates if the observed event rates match the expected event rates in population subgroups. By this analysis, data are grouped by ordering the predicted probabilities and forming the number of groups. The small *p*-value indicates mean that the fit of the model is poor, however, large *p*-values don’t necessarily mean that the model is a good fit. Changing the number of subgroups by very small amounts may result in wild changes in *p*-values. Therefore, we do not think that Hosmer–Lemeshow calculated by our data could be of any significance, due to the fact that, we analyzed a very large population at a national level.

The real-world epidemiology of AKI in western societies is still a matter of debate. The different figures reported in the medical literature are due to the different clinical settings in which studies have been carried out. Several comorbidities have been reported in the list of risk factors for AKI, including diabetes mellitus, cardiovascular disease, chronic liver disease, cancer, and complex surgery [23]; moreover, in individuals with AKI, mortality appears to be related to the stage of the syndrome [24,25]. AKI is a complex condition related to different etiologies and pathophysiological mechanisms; it is commonly diagnosed in hospitalized patients, and leads to increased morbidity, mortality, and health care costs [26]. Different biomarkers have been investigated in order to predict AKI outcomes [27,28], however, we think that consideration of the clinical history of patients developing AKI could improve risk stratification.

Similar to our study, Hsu et al. [29] evaluated dialysis-requiring AKI cases using ICD-9-CM codes. Over a decade (2000–2009), incidence of dialysis-requiring AKI increased, and the risk factors for its development were older age, male sex, black race, and sepsis, as well as acute heart failure, and invasive procedures; mortality was associated with dialysis-requiring AKI and showed an increased frequency [29].

We detected an increasing number of hospitalizations with AKI during the study period. Similar findings were reported by a study analyzing administrative databases between 1988 and 2002. During this period, the incidence of AKI increased from 61 to 288 per 100,000; the incidence of AKI requiring dialysis rose from 4 to 27 per 100,000. On the contrary, IHM decreased from 40.4% to 20.3% in patients with AKI and from 41.3% to 28.1% in those with AKI that required dialysis [30]. The increasing number of admissions and deaths during the study period would be ascribed to two different factors, the first one could be related to a better way of coding by physicians, the second one, could be related to the aging of the population and the change in the organization of Italian health system.

Similar to our study, Liangos et al. [31] defined the epidemiology of AKI using the United States national administrative ICD-9-CM database. AKI was frequent in older, male, black individuals and was associated with chronic kidney disease, congestive heart failure, chronic lung disease, sepsis, and cardiac surgery. In our study, diagnosis of chronic kidney disease was higher in the lowest tertile of the comorbidity score, the same group with a higher prevalence of dialysis-dependent AKI. We cannot exclude the possibility that this group included elderly patients with a low comorbidity burden with decreasing renal function who needed dialysis treatment. Moreover, AKI can be a post-surgical complication, but even in these cases, comorbidity appears to be crucial in determining patients’ prognosis, as shown by Thakar et al. [32] in their study on open-heart surgery. Female gender, congestive heart failure with low ejection fraction, preoperative use of an intra-aortic balloon pump, chronic obstructive pulmonary disease, diabetes, previous surgery, and serum creatinine were associated with mortality and AKI. As in our study, the authors classified subjects in different risk categories of increasing severity [32].

Several studies stratified populations with AKI using different scores, but these indexes were mainly developed in intensive care settings. On the contrary, we tried to stratify our population using a simple score based on administrative data and derived from internal medicine admitted patients. The use of the APACHE II score, especially when modified by the presence or absence of oliguria, should help in predicting the outcome when evaluating interventions for patients with AKI [33]. In 2000, Fiaccadori et al. [34] tested the predictive ability of three general prognostic models: version II of the Acute Physiology and Chronic Health Evaluation (APACHE II), version II of the Simplified Acute Physiology Score, and version II of the Mortality Probability Model at 24 h in a prospective, single-center cohort of patients with AKI in a nephrology setting. They concluded that the APACHE II score was slightly better in predicting mortality. Again, the score used in this study was simple, taking into consideration a small amount of demographic data; in fact, the score included only age and gender; we did not consider race. In Italy, the great majority of hospitalized patients were Caucasian. In 2006, Xue et al. [35] determined the incidence and mortality of AKI in Medicare beneficiaries. They found that older age, male sex, and black race were associated with development of AKI. IHM was 15.2% if AKI was the principal diagnosis at the time of discharge, and 32.6% in discharges with AKI as a secondary diagnosis [35].

Even small acute changes in serum creatinine levels in hospitalized patients increased short-term mortality [36]. In our study, we included elderly patients with AKI without considering different hospital settings. AKI is encountered in 5%–10% of hospitalized patients and up to 60% in individuals admitted to the intensive care unit [37]. In 1996, the epidemiology of AKI was evaluated in 13 tertiary-care Spanish hospitals, covering a population of 4.2 million persons. During a nine-month period, the incidence of AKI was 209 cases per million. Mortality was calculated to be 45%; however, negative outcome in many cases was attributed to underlying diseases, reducing the mortality caused by AKI to 26.7%. Dialysis-dependent AKI was recorded in 36% of individuals [37]. Our mortality was similar, but the percentage of dialysis-dependent AKI was much lower, suggesting that a different selection of the population was investigated. Our data are in agreement with results reported by Chertow et al. [38] who showed that dialysis-dependent AKI and sepsis were independently associated with IHM.

### Limitations

We are aware of several limitations of our study: (i) The design was retrospective, based on administrative data, and we could not assess complications due to AKI such as fluid overload, electrolyte abnormalities, and coagulopathy. It has been reported that complications related to AKI such as fluid overload could be associated with a risk of death [39]. However, our results suggest that conditions diagnosed well before the detection of AKI could be associated with IHM; (ii) we did not provide reasons for admission, specific cause of death, intensive care level, including aggressive therapy and/or device use, neither did we identify AKI on the basis of international Kidney Disease Improving Global Outcomes (KDIGO) guidelines [40]; (iii) we could not evaluate the impact of clinical or biochemical parameters due to the fact that they were not available. There are evident disadvantages in studies based on administrative databases, such as a lack of specific clinical information, the effects of administrative use (i.e., reimbursement), possible misclassification of outcomes, and difficulties in controlling confounding factors [41]. Moreover, as previously stated, race and clinical settings (i.e., surgical and intensive care settings) were not considered; (iv) patients could be wrongly coded, and subjects that are nowadays coded with AKI may have suffered a less severe renal insult than previously. This could be because awareness of AKI has improved; (v) we did not differentiate patients on the basis of the cause of AKI and the treatment setting, we only considered dialysis treatment. On the other hand, only advanced stages of AKI undergo renal replacement therapy. Furthermore, we could not distinguish if AKI happened before or during admission. In 2006, Waikar et al. [42] calculated the performance of ICD-9-CM for acute renal failure and found that the sensitivity was 35.4%, specificity 97.7%, positive predictive value 47.9%, and negative predictive value 96.1% [42]. The validity of AKI codes in administrative databases was analyzed in 2011 in Canada: sensitivity was poor, the median value was 29% (in the range 1%–81%), and median of the positive predictive value was 67% (in the range 15%–96%) [43]. In 2013, Tomlinson et al. [44] calculated a positive predictive value for patients with AKI of 95%. In 2014, Grams et al. [45] validated administrative codes for AKI against the KDIGO AKI definition. They calculated a sensitivity of 17.2% if the comparison was the evaluation of serum creatinine criteria, and 11.7% if the comparison was serum creatinine and urine output-based criteria, whilst specificity was >98% in both cases. Sensitivity was significantly higher when they considered a more recent time period and individuals aged ≥65 years. AKI diagnosed by administrative data was related to more severe disease and higher in-hospital mortality [45]. All these data seem to reinforce our results.

Our study also has some strengths: (i) The high number of records derived from a national administrative database, recording all real diagnoses of AKI; (ii) the long period considered; (iii) the choice of IHM as hard outcome indicator.

## 5. Conclusions

Nowadays, multi-morbidity is receiving greater attention [15], and our findings confirm that comorbidity stratification is crucial to understanding the reasons for IHM of hospitalized elderly patients with AKI. The results of this study emphasize that in elderly subjects, IHM is associated with a degree of renal impairment (especially if the damage needs dialysis treatment), sepsis development, and an increasing burden of comorbidity. Increasing comorbidity score, ascribed to cardiovascular and liver disease, cachexia and cancer, diagnosis of sepsis and advanced renal damage requiring dialysis treatment should be taken into account when evaluating the risk of IHM in hospitalized elderly subjects with AKI.

## Figures and Tables

**Figure 1 jcm-08-01371-f001:**
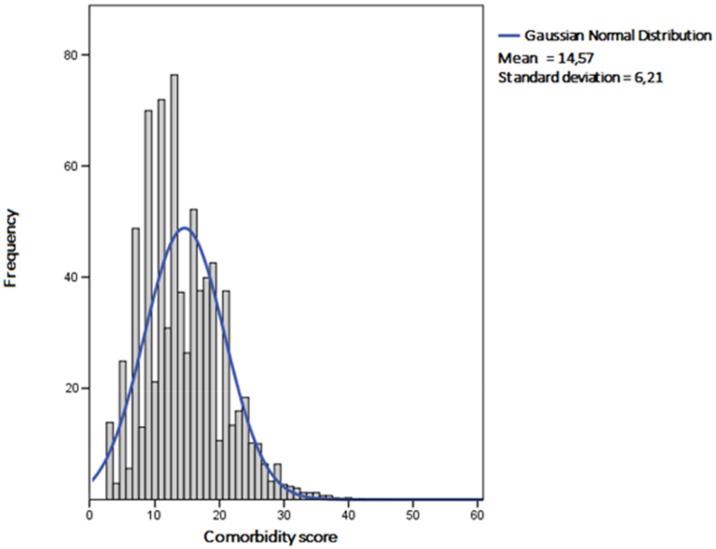
Distribution of the comorbidity score values in the population investigated.

**Figure 2 jcm-08-01371-f002:**
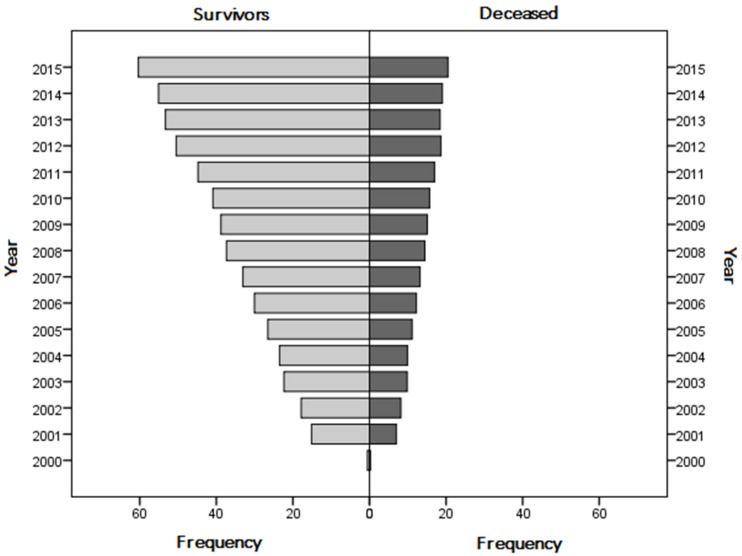
Prevalence of survivors and in-hospital mortality during the study period.

**Figure 3 jcm-08-01371-f003:**
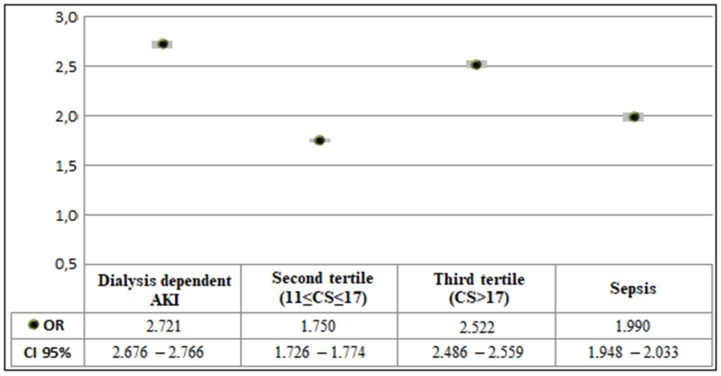
Multivariate analysis results relating the association between in-hospital mortality, comorbidity, and dialysis treatment in individuals with acute kidney injury. CS: Comorbidity Score (in all cases *p* < 0.001; Hosmer and Lemeshow test *p* < 0.001).

**Table 1 jcm-08-01371-t001:** Points assigned to different conditions in order to calculate the score for risk of in-hospital mortality (IHM).

Items	Score
Age 0–60 (years)	0
Age 61–70 (years)	3
Age 71–80 (years)	7
Age 81–90 (years)	11
Age 91+ (years)	16
Chronic kidney disease	1
Male gender	2
Neurological disorders	3
Lymphoma	4
Solid tumor without metastasis	4
Ischemic heart disease	5
Congestive heart failure	5
Coagulopathy	8
Fluid and electrolyte disorders	8
Liver disease	10
Cachexia	11
Metastatic cancer	12

**Table 2 jcm-08-01371-t002:** Characteristics of the considered population with acute kidney injury.

Total Number of Records	760,664
Men, (n (%))	397,174 (52.2)
Women, (n (%))	363,490 (47.8)
Age (years)	80.5 ± 7.8
Comorbidity score	14.57 ± 6.21
Dialysis-dependent acute kidney injury (AKI), (n (%))	68,653 (9)
Diabetes, (n (%))	115,238 (15.6)
Sepsis, (n (%))	39,144 (5.1)
Length-of-stay (LOS) (days)	13.72 ± 15.49
Deceased subjects, (n (%))	210,661 (27.7)

**Table 3 jcm-08-01371-t003:** Comparison of survivors and deceased individuals with acute kidney injury.

Parameters	Survivors	Deceased	*p*
	n = 550,003	n = 210,661	
Men, (n (%))	288,120 (52.4)	109,054 (51.8)	<0.001
Women, (n (%))	261,883 (47.6)	101,607 (48.2)	
Age (years)	80 ± 7.72	81.9 ± 7.9	<0.001
Comorbidity score	14.04 ± 6.02	15.96 ± 6.48	<0.001
Dialysis-dependent AKI, (n (%))	37,598 (6.8)	31,055 (14.7)	<0.001
Diabetes, (n (%))	95,433 (17.4)	22,866 (10.9)	<0.001
Sepsis, (n (%))	23,230 (4.2)	15,914 (7.6)	<0.001

**Table 4 jcm-08-01371-t004:** Comparison of subjects divided into tertiles.

Parameters	First Tertile (Score < 11)	Second Tertile (11 ≤ Score ≤ 17)	Third Tertile (Score > 17)	*p*
Number of subjects	200,131 (26.3%)	332,533 (43.7%)	228,000 (30%)	
Length-of-stay (days)	15.6 ± 19	13.59 ± 14.5	12.28 ± 13.1	<0.001
Comorbidity score	7.48 ± 1.99	13.67 ± 2,04	22.12 ± 4.13	<0.001
Males, (n (%))	119,798 (59.9%)	144,464 (43.4%)	132,912 (58.3%)	<0.001
Females, (n (%))	80,333 (40.1%)	188,069 (56.6%)	95,088 (41.7%)	
Deceased, (n (%))	38,740 (19.4%)	92,987 (28%)	78,934 (34.6%)	<0.001
Dialysis-dependent AKI, (n (%))	26,901 (13.4%)	29,054 (8.7%)	12,698 (5.6%)	<0.001
Sepsis, (n (%))	12,652 (6.3%)	17,944 (5.4%)	8,548 (3.7%)	<0.001
Chronic kidney disease (n (%))	32,217 (16.1%)	49,308 (14.8%)	33,713 (14.8%)	<0.001
Neurological disorders (n (%))	2609 (1.3%)	8661 (2.6%)	6642 (2.9%)	<0.001
Lymphoma (n (%))	2856 (1.4%)	9145 (2.8%)	4442 (1.9%)	<0.001
Solid tumor without metastasis (n (%))	5888 (2.9%)	29,907 (9%)	35,829 (15.7%)	<0.001
Ischemic heart disease (n (%))	2042 (1%)	14,001 (4.2%)	18,900 (8.3%)	<0.001
Congestive heart failure (n (%))	7109 (3.6%)	67,211 (20.2%)	72,039 (31.6%)	<0.001
Coagulopathy (n (%))	0	2780 (0.8%)	5486 (2.4%)	<0.001
Fluid and electrolyte disorders (n (%))	0	33,322 (10%)	93,380 (41%)	<0.001
Liver disease (n (%))	0	8578 (2.6%)	25,437 (11.2%)	<0.001
Cachexia (n (%))	0	441 (0.1%)	14,922 (6.5%)	<0.001
Metastatic cancer (n (%))	0	2009 (0.6%)	28,416 (12.5%)	<0.001

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
