# Peer review of "Acute Kidney Injury and In-Hospital Mortality: A Retrospective Analysis of a Nationwide Administrative Database of Elderly Subjects in Italy"

_jcm, 2019, doi:10.3390/jcm8091371_

Round 1
Reviewer 1 Report
This was a large, database, retrospective cohort study evaluating the association between AKI and in hospital mortality by Fabbian et al. While consisting of a large registry of patients, there are several issues which prevent its publication in its current form.
Major Points
Results - understanding that CKD was not taken into account for the comorbidity calculation, is it possible for the authors to report on pre-existing rates of CKD in their patient population.
Results - is it possible for the authors to comment on the prevalence of diabetes mellitus in their cohort? Also, should this be added to conditions associated with in-hospital risk of mortality? and if not, why?
The authors have access to a 15 year database of patients admitted with a diagnosis of AKI. Is it possible to stratify this data in 1 year increments in order to determine trends in outcomes (i.e., IHM) for these patients? This would greatly add to the impact of this manuscript.
Minor Points
Introduction - would also comment on societal and health economic implications of AKI
Author Response
Dear Editor,
Thank you very much for considering our manuscript (Manuscript ID: jcm-554636) entitled “Acute kidney injury and in-hospital mortality: a nationwide retrospective study in hospitalized elderly subjects in Italy” by Fabio Fabbian, Caterina Savriè, Alfredo De Giorgi, Rosaria Cappadona, Emanuele Di Simone, Benedetta Boari, Alda Storari, Massimo Gallerani, Roberto Manfredini, for possible publication in Journal of Clincal Medicine. We carefully read all reviewers’ concerns and we changed the text accordingly. The whole paper has been re-written, and in our opinion reviewers’ advices have greatly improved the paper. Also the title of the paper has been changed in order to better explain our point. The new title is “Acute kidney injury and in-hospital mortality: a retrospective analysis of a nationwide administrative database of elderly in Italy”. All changes are highlighted in the text
Answer to reviewers
Please also note that we have found some similarity between your manuscript and already published content. This is not allowed in our journal and we kindly ask you to check the report in the attachment and revise the highlighted paragraphs (i.e., lines 33-34, sections 2.1., 2.2., 2.3., lines 209-220, lines 224-229) in order to reduce similarity.
All the sections have been changed starting from introduction, methods, results and discussion. Also abstract and references are now different.
If the reviewers have suggested that your manuscript should undergo extensive English editing, please address this during revision. We suggest that you have your manuscript checked by a native English speaking colleague or use a professional English editing service. Alternatively, MDPI provides an English editing service checking grammar, spelling, punctuation and some improvement of style where necessary for an additional charge (extensive re-writing is not included), see details at https://www.mdpi.com/authors/english.
The English editing has been performed by MDPI service.
Reviewer 1
This was a large, database, retrospective cohort study evaluating the association between AKI and in hospital mortality by Fabbian et al. While consisting of a large registry of patients, there are several issues which prevent its publication in its current form.
Major Points
Results - understanding that CKD was not taken into account for the comorbidity calculation, is it possible for the authors to report on pre-existing rates of CKD in their patient population.
Prevalence of CKD has been added and discussed in the new version of the paper
Results - is it possible for the authors to comment on the prevalence of diabetes mellitus in their cohort? Also, should this be added to conditions associated with in-hospital risk of mortality? and if not, why?
Prevalence of diabetes has been added and discussed in the new version of the paper
The authors have access to a 15 year database of patients admitted with a diagnosis of AKI. Is it possible to stratify this data in 1 year increments in order to determine trends in outcomes (i.e., IHM) for these patients? This would greatly add to the impact of this manuscript.
A graph reporting these data has been added in the results section.
Minor Points
Introduction - would also comment on societal and health economic implications of AKI
A few words and a reference about economic impact of AKI has been added in the introduction section.
Reviewer 2 Report
Dear colleagues,
Thank you for having me review this manuscript. The paper has its merits. The authors put their nationwide electronic healthcare database to good use. The cohort has a multi-center representation. The number of AKI-dialysis patients is large (> 68,000), allowing meaningful analysis.
My chief criticism is that the study aim/hypothesis is not strong, despite the wealth of data made available. The authors seek to examine if the modified Elixhauser index + AKI-dialysis would predict mortality in a large multi-center AKI “elderly” cohort. There is nothing new about this concept. There is a lack of novelty. The results are too brief, and may add little interest to the reader.
There is also a lack of clarity regarding the age cut-off in the study criteria. (What do the authors mean by “elderly AKI”? The mean comorbid score is only 6 (but age group > 70 years already contribute 7 to 11 points to the score); how could it be that the average comorbid score be so low? This needs to be clarified.
Major suggestions:
1. I suggest a major change to the study aims and hypothesis. Would the authors be willing to take further steps to examine, the outcomes of dialysis in AKI patients with increasing comorbidity index and age? There may not be a strong benefit to dialyze AKI patients with very high comorbid burden. How would the %death or hospital days compare between those who undergo dialysis versus none in the very high comorbid group? To do so, the authors need to breakdown the cohort by percentiles in comorbid score and examine the outcomes between dialysis and no dialysis in different tiers. This will be more informative and more relevant to our ageing population with multiple chronic diseases. For all we know, the difference in mortality between dialysis and no dialysis would gradually reduce with higher tiers of comorbid burden.
Minor suggestions:
2. The paper is on AKI. AKI data needs to be interpreted along with information on critical illness / infections / pneumonias / sepsis. Majority of in hospital AKI is related to sepsis. It would be preferred if there are disease codes for the above that the authors could include in the paper, and that would help improve the multivariate analysis.
3. It would be good to understand the hospital days (again, is this available in their electronic data?); if they could show us, that despite dialysis and extended hospital days, mortality remains high in patients with very high comorbid score. That would send a strong message to healthcare providers to consider de-escalating complex care in an elderly patient with advanced comorbidities.
4. The discussion needs to be centered on the study’s results/findings and the relevance to practice. The authors have effectively turned it into a literature review, which is not the intent of an original article.
I see value in the paper and I hope the authors could work on these suggestions and optimise the potential of their very rich patient database. Thank you for advancing healthcare knowledge!
Author Response
Dear Editor,
Thank you very much for considering our manuscript (Manuscript ID: jcm-554636) entitled “Acute kidney injury and in-hospital mortality: a nationwide retrospective study in hospitalized elderly subjects in Italy” by Fabio Fabbian, Caterina Savriè, Alfredo De Giorgi, Rosaria Cappadona, Emanuele Di Simone, Benedetta Boari, Alda Storari, Massimo Gallerani, Roberto Manfredini, for possible publication in Journal of Clincal Medicine. We carefully read all reviewers’ concerns and we changed the text accordingly. The whole paper has been re-written, and in our opinion reviewers’ advices have greatly improved the paper. Also the title of the paper has been changed in order to better explain our point. The new title is “Acute kidney injury and in-hospital mortality: a retrospective analysis of a nationwide administrative database of elderly in Italy”. All changes are highlighted in the text
Answer to reviewers
Please also note that we have found some similarity between your manuscript and already published content. This is not allowed in our journal and we kindly ask you to check the report in the attachment and revise the highlighted paragraphs (i.e., lines 33-34, sections 2.1., 2.2., 2.3., lines 209-220, lines 224-229) in order to reduce similarity.
All the sections have been changed starting from introduction, methods, results and discussion. Also abstract and references are now different.
If the reviewers have suggested that your manuscript should undergo extensive English editing, please address this during revision. We suggest that you have your manuscript checked by a native English speaking colleague or use a professional English editing service. Alternatively, MDPI provides an English editing service checking grammar, spelling, punctuation and some improvement of style where necessary for an additional charge (extensive re-writing is not included), see details at https://www.mdpi.com/authors/english.
The English editing has been performed by MDPI service.
Reviewer 2
Dear colleagues,
Thank you for having me review this manuscript. The paper has its merits. The authors put their nationwide electronic healthcare database to good use. The cohort has a multi-center representation. The number of AKI-dialysis patients is large (> 68,000), allowing meaningful analysis.
Thank you for your kind words.
My chief criticism is that the study aim/hypothesis is not strong, despite the wealth of data made available. The authors seek to examine if the modified Elixhauser index + AKI-dialysis would predict mortality in a large multi-center AKI “elderly” cohort. There is nothing new about this concept. There is a lack of novelty. The results are too brief, and may add little interest to the reader.
The results section has been enlarged adding new data in order to capture readers’ interest.
There is also a lack of clarity regarding the age cut-off in the study criteria. (What do the authors mean by “elderly AKI”? The mean comorbid score is only 6 (but age group > 70 years already contribute 7 to 11 points to the score); how could it be that the average comorbid score be so low? This needs to be clarified.
Thank you for your comments, in the new version of the paper the reviewer could find recalculation of the score, unfortunately in the previous version there were typos errors. Hopefully the reviewer could find the new version of the paper clearer.
Major suggestions:
I suggest a major change to the study aims and hypothesis. Would the authors be willing to take further steps to examine, the outcomes of dialysis in AKI patients with increasing comorbidity index and age? There may not be a strong benefit to dialyze AKI patients with very high comorbid burden. How would the %death or hospital days compare between those who undergo dialysis versus none in the very high comorbid group? To do so, the authors need to breakdown the cohort by percentiles in comorbid score and examine the outcomes between dialysis and no dialysis in different tiers. This will be more informative and more relevant to our ageing population with multiple chronic diseases. For all we know, the difference in mortality between dialysis and no dialysis would gradually reduce with higher tiers of comorbid burden.Thank you for your comments, in the new version of the paper the results section of the paper has been improved and we tried to answer to all criticisms correcting previous tables and figure and adding a new table and new figures. In the new version figures are 3 and tables are 4.
Minor suggestions:
The paper is on AKI. AKI data needs to be interpreted along with information on critical illness / infections / pneumonias / sepsis. Majority of in hospital AKI is related to sepsis. It would be preferred if there are disease codes for the above that the authors could include in the paper, and that would help improve the multivariate analysis.Diagnosis of sepsis has been added to our analysis As reviewer suggested, sepsis was independently associated with in-hospital mortality, although its diagnosis was higher in the first tertile of comorbidity burden. These data have been added in the results section and discussed.
It would be good to understand the hospital days (again, is this available in their electronic data?); if they could show us, that despite dialysis and extended hospital days, mortality remains high in patients with very high comorbid score. That would send a strong message to healthcare providers to consider de-escalating complex care in an elderly patient with advanced comorbidities.Thank you again for this suggestion, length of hospital stay has been calculated and related to comorbidity burden and dialysis dependent AKI.
The discussion needs to be centered on the study’s results/findings and the relevance to practice. The authors have effectively turned it into a literature review, which is not the intent of an original article.Discussion has been changed accordingly.
I see value in the paper and I hope the authors could work on these suggestions and optimise the potential of their very rich patient database. Thank you for advancing healthcare knowledge!
Thank you for your kind words.
Reviewer 3 Report
The authors conducted a potentially interesting study which shows the possible usefulness of big data analysis in medicine. Nevertheless, there are many aspects to clarify with the aim to facilitate a better understanding.
GLOBAL COMMENTS:
-A revision of English language use is necessary, there are many confusing expressions throughout the text.
-The main objective of the study should by clarified, starting with the title. Was the real main objetive the validation of the "Elixhauser score"?
ABSTRACT: change morbidity to mortality in the second line.
INTRODUCTION: must be deeply revised, it is not consistent with the rest of the manuscript. For example, the authors do not mention the "Elixhauser index" in the introduction, which is a capital tool during the development of the study.
EXPERIMENTAL SECTION: the modification performed to the index by removing renal disease could be considered a significative bias, since patients with previous severe renal disease have a higher risk of complications and mortality than dose with AKI not linked to previous renal disease.
RESULTS: the actual presentation of the results is very confusing, it should be corrected. The univariate analysis is not clear, were the variables included in the comorbidity score also analyzed? the authors should explain if there were any difference in the distribution of those variables between two group (deceased and survivors).
Table 3: the differences regarding gender and age between the two groups are very little and it is difficult to believe a P<0.001 in those cases, the authors should explain the statistical methodology which leaded to those results or provide the database in order to favor transparency.
Multivariable analysis: which variables were included in the analysis? which was the multivariable analysis used? was the goodness of fit measured? by which method? this kind of analysis should be clear to validate any conclusion of the study.
DISCUSSION: the authors starts the discussion with the affirmation that "the study intended to evaluate [...] the validity of a novel comorbidity score. This is the first moment in the manuscript that they talk about this objective, it must be added in the introduction and experimental sections. In any case, the study does not seem to be designed to this purpose, and the methodology should be revised. The validation can not be stablished if the only comparator if the need of dialysis and the authors can not affirm that this score "determines the risk of IHM" in patients with AKI.
The discussion section is more a literature review than a real discussion, must be deeply revised, especially the section starting in line 203 page 6, in which the authors do not analyze or compare any result and, as it is actually written, can be deleted.
CONCLUSIONS: if the authors modify the previous points, the conclusions could be probably also modified.
Author Response
Dear Editor,
Thank you very much for considering our manuscript (Manuscript ID: jcm-554636) entitled “Acute kidney injury and in-hospital mortality: a nationwide retrospective study in hospitalized elderly subjects in Italy” by Fabio Fabbian, Caterina Savriè, Alfredo De Giorgi, Rosaria Cappadona, Emanuele Di Simone, Benedetta Boari, Alda Storari, Massimo Gallerani, Roberto Manfredini, for possible publication in Journal of Clincal Medicine. We carefully read all reviewers’ concerns and we changed the text accordingly. The whole paper has been re-written, and in our opinion reviewers’ advices have greatly improved the paper. Also the title of the paper has been changed in order to better explain our point. The new title is “Acute kidney injury and in-hospital mortality: a retrospective analysis of a nationwide administrative database of elderly subjects in Italy”. All changes are highlighted in the text
Answer to reviewers
Please also note that we have found some similarity between your manuscript and already published content. This is not allowed in our journal and we kindly ask you to check the report in the attachment and revise the highlighted paragraphs (i.e., lines 33-34, sections 2.1., 2.2., 2.3., lines 209-220, lines 224-229) in order to reduce similarity.
All the sections have been changed starting from introduction, methods, results and discussion. Also abstract and references are now different.
If the reviewers have suggested that your manuscript should undergo extensive English editing, please address this during revision. We suggest that you have your manuscript checked by a native English speaking colleague or use a professional English editing service. Alternatively, MDPI provides an English editing service checking grammar, spelling, punctuation and some improvement of style where necessary for an additional charge (extensive re-writing is not included), see details at https://www.mdpi.com/authors/english.
The English editing has been performed by MDPI service.
Reviewer 3
The authors conducted a potentially interesting study which shows the possible usefulness of big data analysis in medicine. Nevertheless, there are many aspects to clarify with the aim to facilitate a better understanding.
GLOBAL COMMENTS:
-A revision of English language use is necessary, there are many confusing expressions throughout the text.
The English editing has been performed by MDPI service.
-The main objective of the study should by clarified, starting with the title. Was the real main objetive the validation of the "Elixhauser score"?
Thank you for your suggestion, the text has been changed accordingly.
ABSTRACT: change morbidity to mortality in the second line.
The text has been changed accordingly.
INTRODUCTION: must be deeply revised, it is not consistent with the rest of the manuscript. For example, the authors do not mention the "Elixhauser index" in the introduction, which is a capital tool during the development of the study.
The text has been changed accordingly.
EXPERIMENTAL SECTION: the modification performed to the index by removing renal disease could be considered a significative bias, since patients with previous severe renal disease have a higher risk of complications and mortality than dose with AKI not linked to previous renal disease.
The score has been calculated again considering chronic kidney disease, moreover more information has been added in the results section. In the new version of the paper the results section of the paper has been improved and we tried to answer to all criticisms correcting previous tables and figure and adding a new table and new figures. In the new version figures are 3 and tables are 4.
RESULTS: the actual presentation of the results is very confusing, it should be corrected. The univariate analysis is not clear, were the variables included in the comorbidity score also analyzed? the authors should explain if there were any difference in the distribution of those variables between two group (deceased and survivors).
In the new version results section is completely different, the score of the whole population has been classified in tertiles and the groups have been compared, showing important differences.
Table 3: the differences regarding gender and age between the two groups are very little and it is difficult to believe a P<0.001 in those cases, the authors should explain the statistical methodology which leaded to those results or provide the database in order to favor transparency.
We checked all calculations, and the differences were confirmed. However, our database could be available on request.
Multivariable analysis: which variables were included in the analysis? which was the multivariable analysis used? was the goodness of fit measured? by which method? this kind of analysis should be clear to validate any conclusion of the study.
Multivariate analysis has been performed again and goodness of fit is now reported under figure 3.
DISCUSSION: the authors starts the discussion with the affirmation that "the study intended to evaluate [...] the validity of a novel comorbidity score. This is the first moment in the manuscript that they talk about this objective, it must be added in the introduction and experimental sections. In any case, the study does not seem to be designed to this purpose, and the methodology should be revised. The validation can not be stablished if the only comparator if the need of dialysis and the authors can not affirm that this score "determines the risk of IHM" in patients with AKI.
The discussion section is more a literature review than a real discussion, must be deeply revised, especially the section starting in line 203 page 6, in which the authors do not analyze or compare any result and, as it is actually written, can be deleted.
Thank you for your suggestions, discussion section has been written again.
CONCLUSIONS: if the authors modify the previous points, the conclusions could be probably also modified.
Conclusion section has been modified.
We hope reviewers could appreciate our work on the previous version. The new version of the paper appears as a new one, also title was modified in order to be more attractive for readers.
Thank you again for all your suggestions.
Best regards
Fabio Fabbian on the behalf of all authors
Round 2
Reviewer 1 Report
Fabbian et al. have greatly re-worked their manuscript, data analysis, statistical methods and results presentation in their revised manuscript. It is now much improved when compared to the earlier version of their manuscript.
However, one aspect that requires minor attention is Figure 2. Prevalence of survivors and in-hospital mortality during the study period. At this time, I am unsure how to interpret this figure. Furthermore, while being discussed on lines 234-238, specific mention to the findings in this study would warrant exploration in the discussion. This would also greatly serve to explain the figure and make it more manageable to readers.
Otherwise, the manuscript is much improved and would be suitable to publication.
Author Response
Thank you for your kind words. You are right, and after examination of data we added the following sentence in the discussion section “The increasing number of admissions and deaths during the study period would be ascribed to two different factors, the first one could be related to a better way of coding by physicians, the second one, could be related to the aging of the population and the change in the organization of Italian health system”.
Reviewer 2 Report
Thank you for the extensive revision made and graciously accepting the suggestions.
This paper highlights the changing landscape and epidemiology of hospital-AKI in modern-day medicine. It provides hospital-wide data beyond the confines of critical care and highlights the association between multi-morbidity, advanced cardiovascular diseases, and cancer on AKI and mortality. The results might have an impact on healthcare resource utilization and physician threshold to offer dialysis in the elderly with a very high comorbid burden. I would support this article for publication. Thank you.
Author Response
Thank you very much for your comment.
Reviewer 3 Report
The authors have improved the manuscript, but still have to clarify some aspects:
Methodology:
The authors say that the variables included in the logistic regression analysis were only tertiles of the comorbidity score, dialysis and sepsis. They should explain the reason for this, since there are other variables with statistical significance in the univariate analysis.
Was the logistic regression analysis adjusted by age and sex?
Results:
The division into tertiles could be useful to clarify the results but the comparisons are still doubtful, how could be possible a P<0.001 in every comparison? Which test did the authors use for the comparisons reflected in table 4? (note that most of these analysis are 3x2 tables).
Logistic regression multivariable analysis: if Hosmer-Lemeshow (please correct the name of the test) was really aplied and its result is P<0.001 as the authors say there are only 2 posibilities:
1.This is the real P, which means that the model is completely unadjusted and not valid, it should be adjusted if possible, if not possible it should be deleted because this analysis can not be accepted.
2.The authors made a mistake and this is not the real P of the Hosmer-Lemeshow test.
Discussion:
It is still chaotic, it should be rewritten, the main topics to analyse are only 3, I suggest this order:
1.General characteristics of patients with AKI.
2.In-hospital mortality of patients with AKI.
3.Main factors linked to in-hospital mortality in patients with AKI.
Other aspects and data are completely irrelevant and should be deleted.
Conclusions:
Which are the conclusions of the authors? they should be clear and concise, conclusions are not comparisons with other studies.
Author Response
The authors have improved the manuscript, but still have to clarify some aspects:
Methodology:
The authors say that the variables included in the logistic regression analysis were only tertiles of the comorbidity score, dialysis and sepsis. They should explain the reason for this, since there are other variables with statistical significance in the univariate analysis.
The reason for dividing the population in tertiles and adding sepsis to the logistic regression is due to the fact that, firstly it is a reviewer suggestion, secondly that sepsis is a well-known risk factor for mortality in patients suffering acute kidney injury, moreover such an item is not included in the score. The division in tertiles could better explain to the readers the importance of comorbidity burden.
Was the logistic regression analysis adjusted by age and sex?
We could not adjust regression analysis for gender and age, because the two items are included in the score. An adjustment for gender and age in the logistic regression could alter the results.
Results:
The division into tertiles could be useful to clarify the results but the comparisons are still doubtful, how could be possible a P<0.001 in every comparison? Which test did the authors use for the comparisons reflected in table 4? (note that most of these analysis are 3x2 tables).
Logistic regression multivariable analysis: if Hosmer-Lemeshow (please correct the name of the test) was really aplied and its result is P<0.001 as the authors say there are only 2 posibilities:
This is the real P, which means that the model is completely unadjusted and not valid, it should be adjusted if possible, if not possible it should be deleted because this analysis can not be accepted. The authors made a mistake and this is not the real P of the Hosmer-Lemeshow test.Hosmer and Lemeshon has been changed with Hosmer-Lemeshow, we are sorry for this mistake. The reason for obtaining in all cases p<0.001 could be due to the very large sample of patients analyzed. In table 4 we used contingency table (method runs by SPSS). Finally, as previously requested by reviewer we reported the goodness of fit test (result obtained by SPSS). As suggested by reviewer, the Hosmer-Lemeshow test is a goodness of fit test for logistic regression, especially for risk prediction models. The goodness of fit test tells you how well your data fits the model. Specifically, the Hosmer-Lemeshow test calculates if the observed event rates match the expected event rates in population subgroups, and the test, is only used for binary response variables. This test, is usually run using SPSS and data are first regrouped by ordering the predicted probabilities and forming the number of groups. The output returns a chi-square value (a Hosmer-Lemeshow chi-squared) and a p-value. As suggested by reviewer, small p-values mean that the model is a poor fit. Like most goodness of fit tests, these small p-values mean that the model is not a good fit. However, large p-values don’t necessarily mean that the model is a poor fit, just that there isn’t enough evidence to say it’s a good fit. Many situations can cause large p-values, including poor test power. Several problems have been identified with the Hosmer-Lemeshow test. For example, it doesn’t take overfitting into account and tends to have low power. There is also very little guidance to selecting the number of subgroups. Changing the number of subgroups by very small amounts may result in wild changes in p-values. As such, the selection for subgroups is often confusing and arbitrary. Therefore, the test is usually not recommended. Taking all this into account, a sentence regarding the Hosmer-Lemeshow test has been added in the discussion section.
“We run the Hosmer-Lemeshow test in order to evaluate goodness of fit of our logistic regression, and we could detect that our data were not fitting well into the model. The Hosmer-Lemeshow test calculates if the observed event rates match the expected event rates in population subgroups. By this analysis, data are grouped by ordering the predicted probabilities and forming the number of groups. The small p-value indicates mean that the fit of the model is poor, however, large p-values don’t necessarily mean that the model is a good fit. Changing the number of subgroups by very small amounts may result in wild changes in p-values. Therefore, we do not think that Hosmer-Lemeshow calculated by our data could be of any significance, due to the fact that, we analyzed a very large population at a national level.”
Discussion:
It is still chaotic, it should be rewritten, the main topics to analyse are only 3, I suggest this order:
General characteristics of patients with AKI. In-hospital mortality of patients with AKI. Main factors linked to in-hospital mortality in patients with AKI.Other aspects and data are completely irrelevant and should be deleted.
Conclusions:
Which are the conclusions of the authors? They should be clear and concise, conclusions are not comparisons with other studies.
Discussion and conclusions have been changed following reviewer’s suggestion, however we think that any data could be considered relevant for discussing our results. Obviously, sequence of references has been changed accordingly.
Round 3
Reviewer 3 Report
The authors have improved adequately the manuscript and explained all doubts regarding methodology.